# Use of Systemic Antibiotics in Patients with COVID-19 in Colombia: A Cross-Sectional Study

**DOI:** 10.3390/antibiotics12020252

**Published:** 2023-01-26

**Authors:** Luis Fernando Valladales-Restrepo, Ana Camila Delgado-Araujo, Luisa Fernanda Echeverri-Martínez, Verónica Sánchez-Ríos, Jorge Enrique Machado-Alba

**Affiliations:** 1Grupo de Investigación en Farmacoepidemiología y Farmacovigilancia, Universidad Tecnológica de Pereira-Audifarma S.A., Pereira 660002, Colombia; 2Grupo de Investigación Biomedicina, Facultad de Medicina, Fundación Universitaria Autónoma de las Américas, Pereira 660005, Colombia; 3Semillero de Investigación en Farmacología Geriátrica, Facultad de Medicina, Fundación Universitaria Autónoma de las Américas, Pereira 660005, Colombia

**Keywords:** COVID-19, SARS-CoV-2, inappropriate prescribing, antibacterial agents, intensive care units, Colombia

## Abstract

Antibiotics are frequently prescribed to patients with COVID-19. The aim was to determine the pattern of use of systemic antibiotics in a group of patients diagnosed with COVID-19 in Colombia between 2020–2022. This was a descriptive cross-sectional study designed to identify antibiotics prescription patterns for patients diagnosed with COVID-19 treated in eight clinics in Colombia. The AWaRe tool of the World Health Organization (WHO) was used to classify the antibiotics. A total of 10,916 patients were included. The median age was 57 years, and 56.4% were male. A total of 57.5% received antibiotics, especially ampicillin/sulbactam (58.8%) and clarithromycin (47.9%). Most of the antibiotics were classified as Watch (65.1%), followed by Access (32.6%) and Reserve (2.4%). Men (OR: 1.29; 95%CI: 1.17–1.43), older adults (OR: 1.67; 95%CI: 1.48–1.88), patients with dyspnea (OR: 1.26; 95%CI: 1.13–1.41), rheumatoid arthritis (OR: 1.94; 95%CI: 1.17–3.20), and high blood pressure at admission (OR: 1.45; 95%CI: 1.29–1.63), patients treated in-hospital (OR: 5.15; 95%CI: 4.59–5.77), patients admitted to the ICU (OR: 10.48; 95%CI: 8.82–12.45), patients treated with systemic glucocorticoids (OR: 3.60; 95%CI: 3.21–4.03) and vasopressors (OR: 2.10; 95%CI: 1.60–2.75), and patients who received invasive mechanical ventilation (OR: 2.37; 95%CI: 1.82–3.09) were more likely to receive a systemic antibiotic. Most of the patients diagnosed with COVID-19 received antibiotics, despite evidence showing that bacterial coinfection is rare. Antibiotics from the Watch group predominated, a practice that goes against WHO recommendations.

## 1. Introduction

Coronavirus disease 2019 (COVID-19), caused by severe acute respiratory syndrome virus type-2 (SARS-CoV-2), was declared a pandemic by the World Health Organization (WHO) in March 2020 [1]. As of 1 November 2022, worldwide, more than 630 million people have been infected, and more than 6.5 million people have died [2]. In Colombia, according to data from the National Institute of Health, more than 6.3 million cases have been reported, and 141 thousand people have died [3]. Most COVID-19 patients have mild or moderate illness [4,5,6], but some require hospitalization for serious respiratory complications, sepsis, acute renal failure, and disseminated intravascular coagulation, among other issues [7].

Antibiotics have been used in hospitals to empirically treat patients with suspected COVID-19, probably because the clinical and radiological characteristics of viral and bacterial pneumonia overlap [8] and initially, some studies (with inadequate methodological designs) showed benefits [9]. The use of antibiotics should ideally be reserved for patients with clinical suspicion or with paraclinical confirmation of bacterial coinfection [4,5,6]. However, an appreciable prescription of antibiotics has been documented among hospitalized patients with COVID-19 [10,11,12], and the problem is much greater in low- and middle-income countries [12]. The high use of antibiotics does not correlate with the relatively low prevalence of bacterial coinfection in patients with COVID-19 [10,11,13].

The improper and excessive use of antimicrobials is the main factor that determines the appearance of drug-resistant pathogens [14]. Therefore, the unnecessary prescription of antimicrobials to patients with COVID-19 is a major global concern due to the increased risk of antibiotic resistance, which will lead to increased morbidity, mortality, and health care and social costs [15]. Antimicrobial resistance will likely become an even more significant challenge in the post-COVID-19 pandemic era [15]. The WHO developed an instrument that seeks to improve the quality of antibiotic prescriptions to decrease the spread of resistant microorganisms and reduce adverse reactions and costs. The AWaRe tool classifies antibiotics into three groups: Access, Watch, and Reserve (Appendix A). Access antibiotics are those that should initially be used for the most common and severe infections, are narrow spectrum, and are less expensive; Watch antibiotics should be used in moderation due to the relatively high risk of resistant strains; and Reserve antibiotics are to be used for the treatment of infections by microorganisms resistant to multiple antibiotics [16].

In Colombia, a low- to middle-income country, there is limited information available that addresses this problem [17]. The country’s health system offers universal coverage to the entire population through two affiliation regimes, i.e., contributory, for which costs are paid by workers and employers, and subsidized, i.e., which insures people without the ability to pay, and includes a benefit plan that involves a significant number of systemic antibiotics and antifungals. The objective of this study was to determine the pattern of use of systemic antibiotics in a group of patients diagnosed with COVID-19 in Colombia between 2020–2022.

## 2. Results

A total of 10,916 patients from 181 different cities were included; 56.4% (n = 6155) were male, and the median age was 57.0 years (interquartile range: 41.0–70.0 years). A total of 2.6% (n = 281) were under 18 years of age, 19.9% (n = 2174) were between 18–39 years of age, 40.3% (n = 4395) were between 40–64 years of age, and 34.3% (n = 3748) were 65 years or older; for 2.9% (n = 318), age was unknown. The majority were from the Pacific (n = 4468, 40.9%) and Central (n = 3052, 28.0%) regions. A total of 62.0% (n = 6770) had some chronic comorbidity, the most frequent being high blood pressure (n = 5789; 53.0%) and diabetes mellitus (n = 2347; 21.5%).

The most common clinical manifestations for which the patients consulted were cough (n = 5766; 52.8%), fever (n = 4708; 43.1%), and dyspnea (n = 3712; 34.0%). Most of the patients required medical attention in general hospitalization wards (n = 4042; 37.0%), followed by emergency services (n = 3635; 33.3%) and intensive care units (ICUs) (n = 3239; 29.7%). A total of 62.5% of the patients (n = 6822) required supplemental oxygen, and 20.2% (n = 2208) required invasive mechanical ventilation, among whom 4.3% (n = 94/2208) required a tracheostomy. A total of 1903 (17.3%) patients died in-hospital. Table 1 provides comparisons of sociodemographic, clinical, and pharmacological variables with the care service required by patients with COVID-19.

### 2.1. Use of Antimicrobials

A total of 57.5% (n = 6273) of the patients accounted for 15,722 antibiotic prescriptions. The median was 2.0 antibiotics (IQR: 2.0–3.0) for patients who received any of these drugs. According to the WHO AWaRe classification, the majority corresponded to Watch antibiotics (n = 10,232/15,722; 65.1%), followed by Access antibiotics (n = 5119; 32.6%) and Reserve antibiotics (n = 371; 2.4%). The most prescribed therapeutic groups were penicillins (n = 4475/10,916; 41.0%), macrolides (n = 3756; 34.4%), and cephalosporins (n = 2664; 24.4%), which were used mainly in the ICU and in hospitalization services (Table 1). Watch and Reserve antibiotics were used predominantly in the ICU, and Access antibiotics were used in hospital wards and emergency rooms (Table 2).

The main treatment regimens used were ampicillin sulbactam + clarithromycin (n = 1189, 19.0%), ampicillin sulbactam (n = 733, 11.7%), ceftriaxone + clarithromycin (n = 373, 5.9%), ampicillin sulbactam + azithromycin (n = 329; 5.2%), and ceftriaxone (n = 130; 2.1%). A total of 5.8% (n = 634) of the patients were managed with antifungal drugs, highlighting the prescription of fluconazole (n = 381/10,916; 3.5%) and caspofungin (n = 220; 2.0%). A total of 5.7% (n = 622) of all patients received systemic antibiotics and antifungal agents during care.

### 2.2. Multivariate Analysis

Binary logistic regression was adjusted for sociodemographic, clinical, and pharmacological variables. Male patients (OR: 1.29), those aged 65 years and older (OR: 1.67), those from the Caribbean region (OR: 2.46), those with dyspnea at admission (OR: 1.26), and those with rheumatoid arthritis (OR: 1.94) or high blood pressure (OR: 1.45) had a greater probability of receiving a systemic antibiotic, as well as patients diagnosed with COVID-19 in the first 10 months of the pandemic (OR: 2.00), those admitted to the hospital (OR: 5.15) or ICU (OR: 10.48), those who were treated with systemic glucocorticoids (OR: 3.60) or vasopressors/inotropes (OR: 2.10), and those who required mechanical ventilation (OR: 2.37). No variable reduced this risk (Table 3). The findings were similar when the dependent variable in the binary logistic regression was the use of Watch/Reserve antibiotics (Appendix A).

## 3. Discussion

This study allowed the characterization of the use of systemic antibiotics in a group of patients with a confirmed diagnosis of SARS-CoV-2 infection treated in eight highly complex clinics in Colombia. The median age of these patients was similar to that found in other reports (53.0–61.0 years) [10,17,18,19], and there was a global predominance of males, as described in other studies (51.5–69.0%) [10,17,18,20,21,22,23,24]. The most frequently found comorbidities were high blood pressure and diabetes mellitus, findings that are consistent with what has been reported in other studies [17,19,20,21,23,24,25]. Similarly, the symptom profile was consistent with that described in the literature [17,19,24].

A total of 57.5% of the patients received some systemic antibiotics, a finding that is in line with what has been documented in other studies, but with variable proportions [18,20,21,22,23,25,26,27]. The prevalence of antibiotic use was higher in countries such as Pakistan (89.7–93.7%) [20,26], Bangladesh (92.0%) [21], Spain (87.8%) [23], and the USA (76.3%) [22], similar to that in Scotland (63.9%) [27], Colombia (57.2%) [17], Croatia, Italy, Serbia, and Slovenia (52.7%) [25], and lower than that in Mexico (25.3%) [18]. In addition, we found that 5.8% of the patients received systemic antifungals, a percentage that is slightly higher than that reported in other studies (1.6–2.7%) [20,26,27]. The high exposure to antibiotics reported in most studies is striking [20,21,22,23,25,26,27], especially considering that bacterial coinfection in patients with COVID-19 occurs in only between 5.6 and 8.6% of patients who are treated in-hospital [10,11,13] and in 14.0% of patients treated in ICUs [13].

Clinical practice guidelines do not recommend prophylactic antibiotic coverage for patients with COVID-19 [4,5]. Antibiotics are indicated if there is clinical suspicion or laboratory confirmation of bacterial infection [4,5]. Similarly, Colombian guidelines recommend only empirical antibiotic therapy for patients with suspected coinfection by SARS-CoV-2 and bacterial pneumonia [6]. However, it is likely that the extensive empirical use of antibiotics in these patients occurs because the clinical and radiological characteristics of COVID-19 overlap with those of bacterial infections of the respiratory tract [8]. However, receiving antibiotics in the first 48 h after admission increases the risk of developing infections during hospitalization (OR: 11.4; 95% CI: 1.5–85.0; *p*: <0.01) [28] and of presenting more complications in general (OR: 2.1; 95% CI: 1.8–2.4; *p*: <0.001) [29].

According to the AWaRe classification devised by the WHO, the antibiotics that were most used in this group of patients belonged to the Watch group. This finding is consistent with those reported in other countries [20,21,25,26,30]. For example, in Pakistan and Bangladesh, the proportion of Watch antibiotics was very high (93.4% and 79.9%, respectively) [20,21]; however, in several European countries and India, the proportion was slightly lower (69.9% and 52.4%, respectively) [25,30], with the latter being consistent with our findings. The use of Access antibiotics ranged from 2.3% and 27.7% [20,21,30]. The objective proposed by the WHO is to increase the proportion of global antibiotic consumption in the Access group by at least 60% and to reduce the use of antibiotics in the Watch and Reserve groups, which are associated with an increased risk of antimicrobial resistance [16]. However, as documented in previous studies [20,21,26,30], these objectives and recommendations have not been met. A systematic review indicated that antibiotic resistance has increased during the COVID-19 pandemic and that this is due to self-medication and the empirical administration and prescription of antibiotics by general practitioners [31].

Different variables were found to be related to the general prescription of antibiotics and the use of Watch/Reserve antibiotics. Some sociodemographic variables, such as being male and of older age, were associated with a greater probability of receiving these drugs, consistent with other studies [11,18,23]. Similarly, the presence of some comorbidities increased the risk, as evidenced in other reports [18,23,27]. Although it has been described that patients with diabetes mellitus and chronic obstructive pulmonary disease have a greater susceptibility to infections, this study did not find an increase in the use of antibiotics in them. This is probably due to the fact that the patients had adequate control of their underlying pathologies or that they were severe cases [32,33]. The prescription of antibiotics was significantly higher for patients treated in-hospital and even higher for those treated in ICUs, as well as for those who needed mechanical ventilation or vasopressor support, findings that are consistent with several studies that show that their use increases based on COVID-19 severity [11,18,20,21]. A risk relationship was also found in those who were receiving systemic glucocorticoids, a finding that is consistent with other research [34,35]. Similarly, the use of antibiotics predominated in the first months of the COVID-19 pandemic, consistent with what was reported by Ramzan et al. in Pakistan, where the use of antibiotics decreased significantly after the second peak of COVID-19 [20], and with what was found by Murillo-Zamora et al. in Mexico, where for each additional week that passed since the beginning of the pandemic, the likelihood of receiving antibiotics decreased by 2.0% [18].

Some limitations should be considered when interpreting the results of this study because access to medical records was not obtained to identify the severity and complications of COVID-19 or other clinical and paraclinical variables. The concomitant presence of other infectious pathologies could not be determined. Similarly, due to the study design, the relevance of antibiotic therapy could not be established, and the drugs prescribed outside the health system or not delivered by the dispensing company that the patients may have received are unknown. However, the study included a significant number of patients distributed in several main cities of the national territory and managed in different care services (emergency, hospitalization, or ICU).

## 4. Materials and Methods

This was a cross-sectional study of the prescription patterns of systemic antibiotics used for patients treated in hospitals with a confirmed diagnosis of COVID-19; the patients were identified from reports of confirmed cases by polymerase chain reaction (RT–PCR) or antigen tests performed by the Ospedale Group network in eight clinics located in the cities of Armenia, Barranquilla, Bogotá, Cali, Cartagena, Manizales, Pereira, and Popayán in Colombia. These are highly complex clinics that are reference centers and also treat patients referred from other cities.

From this population, patients of any age, gender, and city of residence with a first confirmed diagnosis of SARS-CoV-2 between March 6, 2020, and August 31, 2022, and treated in the emergency department, a general ward, or the ICU, were selected. With this selection, information on the use of medications was obtained through the dispensing company (Audifarma SA, Pereira, Colombia). A database was designed that allowed the following groups of patient variables to be collected:Sociodemographic data: gender, age (<18 years, 18–39 years, 40–64 years, 65 years or older), and city of origin. City of origin was categorized by department based on the regions of Colombia, taking into account the classification of the National Administrative Department of Statistics (Departamento Administrativo Nacional de Estadística—DANE) of Colombia: Caribbean, Central, Bogotá-Cundinamarca, Pacific and Amazonia, and Orinoquia—Oriental;Clinical symptoms (cough, dyspnea, fever, fatigue, odynophagia, chest pain, and asthenia/adynamia, among others), comorbidities (cardiovascular, respiratory, digestive, endocrine, neurological, psychiatric, rheumatological, and oncological), and mortality;Treatment: The management received by the patients was established from the dispensing of the medications.
Place of care: emergency department, general ward or ICU;Supplemental oxygen: oxygen requirement, mechanical ventilation, and need for tracheostomy;Antibiotics: classified by therapeutic group (aminoglycosides, cephalosporins, fluoroquinolones, macrolides, penicillins, and tetracyclines, among others) and by the WHO AWaRe (Access, Watch, and Reserve) classification (Appendix A) [16];Antifungals: azoles (fluconazole, miconazole, voriconazole, posaconazole, itraconazole, and ketoconazole), polyene antibiotics (amphotericin B and nystatin), echinocandins (caspofungin, micafungin, and anidulafungin), and others;Comedications, grouped into the following categories: (a) antidiabetes drugs, (b) antihypertensive and diuretic drugs, (c) lipid-lowering drugs, (d) antiulcer drugs, (e) systemic glucocorticoids, (f) vasopressors and inotropes, (g) anticoagulants, (h) analgesics and anti-inflammatories, and (i) bronchodilators and inhaled glucocorticoids, among others.

The protocol was approved by the Bioethics Committee of the Universidad Tecnológica de Pereira (Technological University of Pereira) in the category of “research without risk” (approval code: 30-070421). The principles of confidentiality of information established by the Declaration of Helsinki were respected.

The data were analyzed using the statistical package SPSS Statistics, version 26.0 for Windows (IBM, Armonk, NY, USA). A descriptive analysis was performed; qualitative variables are presented as frequencies and proportions, and quantitative variables are presented as measures of central tendency and dispersion (medians and interquartile ranges). Quantitative data were compared using the Mann–Whitney U test and categorical data were compared using X^2^ test or Fisher’s exact test. Multivariate binary logistic regression models were developed, which included the associated variables in the bivariate analyses as well as variables with sufficient plausibility or a reported association to identify variables that are associated with the prescription of antibiotics (yes/no) and the prescription of antibiotics from the Watch/Reserve categories (yes/no). A *p* < 0.05 was considered statistically significant.

## 5. Conclusions

With the findings of this study, it can be concluded that more than half of the patients diagnosed with COVID-19 received antibiotics, despite evidence indicating that bacterial coinfection is not frequent. Most of the antibiotics used were from the Watch group, a practice that goes against the WHO recommendations. The high consumption of antibiotics found among patients with COVID-19 requires the implementation of appropriate antimicrobial administration interventions. The administration of antibiotics does not prevent the unfavorable progression of viral pneumonia and does not decrease mortality. Therefore, the use of antibiotics is not recommended without clear clinical suspicion or confirmation of bacterial co-infection. However, more research is needed to better identify patients with bacterial co-infection and to be able to initiate antimicrobial therapy appropriately.

## Figures and Tables

**Table 1 antibiotics-12-00252-t001:** Comparison of sociodemographic, clinical, and pharmacological variables of 10,916 patients with a diagnosis of COVID-19 by the care service in eight highly complex clinics in Colombia.

Variables	Emergency Care		Hospitalization Care		Intensive Care Unit	
	n = 3935	%	n = 4042	%	n = 3229	%
Male	1804	49.6	2271	56.2	2080	64.2
Age, median (Interquartile range)	45.0 (32.0–60.0)		59.0 (44.0–73.0)		64.0 (53.0–74.0)	
<65 years	2964	81.5	2508	62.0	1696	52.4
≥65 years	671	18.5	1534	38.0	1543	47.6
Origin (Region)	-	-	-	-	-	-
Pacific	1088	29.9	1780	44	1600	49.4
Central	946	26	1337	33.1	769	23.7
Caribbean	408	11.2	837	20.7	513	15.8
Bogota-Cundinamarca	1191	32.8	87	2.2	348	10.7
Oriental-Amazon-Orinoquia	2	0.1	1	0.0	9	0.3
Comorbidities	-	-	-	-	-	-
Arterial hypertension	1038	28.6	2168	53.6	2583	79.7
Diabetes mellitus	365	10.0	879	21.7	1103	34.1
Hypothyroidism	160	4.4	349	8.6	298	9.2
Dyslipidemia	167	4.6	298	7.4	177	5.5
Chronic obstructive pulmonary disease	92	2.5	274	6.8	180	5.6
Symptoms	-	-	-	-	-	-
Cough	2301	63.3	1589	39.3	1876	57.9
Fever	1845	50.8	1212	30.0	1651	51.0
Dyspnoea	921	25.3	1114	27.6	1677	51.8
Fatigue	990	27.2	864	21.4	674	20.8
Odynophagia	1136	31.3	580	14.3	417	12.9
Oxygen requirements	-	-	-	-	-	-
Supplemental oxygen	1027	28.3	2753	68.1	3042	93.9
Mechanic ventilation	70	1.9	142	3.5	1996	61.6
Systemic antibiotics	-	-	-	-	-	-
Penicillins	495	13.6	1857	45.9	2123	65.5
Macrolides	484	13.3	1469	36.3	1803	55.7
Cephalosporins	236	6.5	907	22.4	1521	47.0
Glycopeptides	14	0.4	77	1.9	1112	34.3
Carbapenems	12	0.3	77	1.9	1027	31.7
Lincosamides	16	0.4	95	2.4	111	3.4
Aminoglycosics	15	0.4	38	0.9	159	4.9
Tetracyclines	19	0.5	115	2.8	62	1.9
Oxazolidinones	1	0.0	4	0.1	157	4.8
Fluoroquinolones	8	0.2	68	1.7	77	2.4
Systemic antifungals	-	-	-	-	-	-
Fluconazole	3	0.1	47	1.2	331	10.2
Caspofungin	0	0	4	0.1	216	6.7
Nystatin	1	0	53	1.3	81	2.5
Amphotericin B	0	0	1	0	7	0.2
Voriconazole	0	0	2	0	6	0.2
Comedications	-	-	-	-	-	-
Analgesics and anti-inflammatories	2422	66.6	2977	73.7	2391	73.8
Corticosteroids	1590	43.7	3012	74.5	2844	87.8
Anticoagulants	1041	28.6	3200	79.2	3094	95.5
Anti-ulcer	981	27.0	2961	73.3	3005	92.8
Bronchodilators/inhaled corticosteroids	999	27.5	2035	50.3	1650	50.9
Mortality	139	3.8	352	8.7	1412	43.6

**Table 2 antibiotics-12-00252-t002:** AwaRe (Access, Watch, and Reserve) classification of 6273 patients diagnosed with COVID-19 who received systemic antibiotics by the care service in eight highly complex clinics in Colombia.

Variables	Total		Emergency Care		Hospitalization Care		Intensive Care Unit	
	n = 6.273	%	n = 733	%	n = 2.618	%	n = 2.922	%
Access	4502	71.8	529	72.2	1994	76.2	1979	67.7
Ampicillin/sulbactam	3688	58.8	445	60.7	1656	63.3	1587	54.3
Cefazolin	373	5.9	28	3.8	175	6.7	170	5.8
Clindamycin	222	3.5	16	2.2	95	3.6	111	3.8
Doxycycline	196	3.1	19	2.6	115	4.4	62	2.1
Amikacin	167	2.7	10	1.4	22	0.8	135	4.6
Trimethoprim/sulfamethoxazole	104	1.7	2	0.3	34	1.3	68	2.3
Cephalothin	87	1.4	6	0.8	39	1.5	42	1.4
Ampicillin	80	1.3	11	1.5	33	1.3	36	1.2
Oxacillin	74	1.2	0	0.0	10	0.4	64	2.2
Gentamicin	48	0.8	5	0.7	16	0.6	27	0.9
Penicillin G benzathine	22	0.4	14	1.9	3	0.1	5	0.2
Cephalexin	19	0.3	2	0.3	9	0.3	8	0.3
Penicillin G crystalline	18	0.3	6	0.8	7	0.3	5	0.2
Amoxicillin	14	0.2	3	0.4	6	0.2	5	0.2
Cephradine	3	0.0	0	0.0	2	0.1	1	0.0
Nitrofurantoin	3	0.0	0	0.0	2	0.1	1	0.0
Amoxicillin/clavulanate	1	0.0	0	0.0	1	0.0	0	0.0
Watch	5352	85.3	594	81.0	2036	77.8	2722	93.2
Clarithromycin	3005	47.9	423	57.7	1206	46.1	1376	47.1
Cepepime	1306	20.8	41	5.6	205	7.8	1060	36.3
Ceftriaxone	1270	20.2	170	23.2	585	22.3	515	17.6
Vancomycin	1203	19.2	14	1.9	77	2.9	1112	38.1
Piperacillin/tazobactam	1184	18.9	40	5.5	272	10.4	872	29.8
Meropenem	1058	16.9	11	1.5	57	2.2	990	33.9
Azithromycin	875	13.9	62	8.5	350	13.4	463	15.8
Ciprofloxacin	132	2.1	6	0.8	63	2.4	63	2.2
Ertapenem	93	1.5	2	0.3	23	0.9	68	2.3
Erythromycin	50	0.8	0	0.0	3	0.1	47	1.6
Imipenem/cilastatin	19	0.3	0	0.0	3	0.1	16	0.5
Levofloxacin	11	0.2	0	0.0	2	0.1	9	0.3
Moxifloxacin	11	0.2	2	0.3	3	0.1	6	0.2
Rifaximin	8	0.1	0	0.0	3	0.1	5	0.2
Fosfomycin	4	0.1	0	0.0	0	0.0	4	0.1
Cefuroxime	2	0.0	0	0.0	1	0.0	1	0.0
Ceftazidime	1	0.0	0	0.0	0	0.0	1	0.0
Reserve	326	5.2	3	0.4	31	1.2	292	10.0
Linezolid	162	2.6	1	0.1	4	0.2	157	5.4
Aztreonam	113	1.8	2	0.3	23	0.9	88	3.0
Tigecycline	39	0.6	0	0.0	2	0.1	37	1.3
Ceftazidime/avibactam	28	0.4	0	0.0	1	0.0	27	0.9
Colistin	10	0.2	0	0.0	2	0.1	8	0.3
Daptomycin	10	0.2	0	0.0	1	0.0	9	0.3
Ceftaroline	4	0.1	0	0.0	0	0.0	4	0.1
Polymyxin B	4	0.1	0	0.0	0	0.0	4	0.1
Ceftolozan/tazobactam	1	0.0	0	0.0	0	0.0	1	0.0

**Table 3 antibiotics-12-00252-t003:** Binary logistic regression of the variables related to receiving antibiotics in 10,916 patients with a diagnosis of COVID-19 treated in eight highly complex clinics in Colombia.

Variables	Sig.	OR	CI95%	
			Lower	Upper
Male	<0.001	1.296	1.172	1.433
Age ≥ 65 years	<0.001	1.675	1.486	1.888
Origin Caribbean Region	<0.001	2.463	2.140	2.834
Dyspnea on admission	<0.001	1.265	1.130	1.417
Rheumatoid arthritis	0.010	1.941	1.174	3.209
Ischemic heart disease	0.058	0.751	0.558	1.010
Diabetes mellitus	0.206	1.092	0.953	1.251
Chronic obstructive pulmonary disease	0.060	0.802	0.637	1.009
Chronic kidney disease	0.951	0.992	0.759	1.296
Arterial hypertension	<0.001	1.455	1.297	1.633
First 10 months of the pandemic	<0.001	2.000	1.799	2.224
Emergency Care	<0.001	Reference	Reference	Reference
Hospital Care	<0.001	5.153	4.598	5.776
Intensive Care Unit	<0.001	10.482	8.820	12.457
Systemic corticosteroids	<0.001	3.602	3.217	4.033
Vasopressors—inotropes	<0.001	2.104	1.605	2.758
Invasive mechanical ventilation	<0.001	2.375	1.821	3.098

Sig: Statistical significance; OR: Odds Ratio; CI: Confidence interval.

## Data Availability

Protocols.io https://doi.org/10.17504/protocols.io.rm7vzbk5xvx1/v1 (accessed on 14 December 2022).

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
