# Peer review of "Use of Systemic Antibiotics in Patients with COVID-19 in Colombia: A Cross-Sectional Study"

_antibiotics, 2023, doi:10.3390/antibiotics12020252_

Round 1

Reviewer 1 Report

Thank you for inviting me to review this interesting article on antibiotics use in patients with COVID-19. It is well written and in an interesting way, however, there are some remarks I would suggest implementing in order to increase it scientific value:

Abstract

The Abstract contains 330 words -> Please see the Instructions for the authors, the abstract should not be longer than 200 words

Introduction

In the introduction section I would suggest adding some information of first studies that showed antibiotic use in COVID-19 necessary. Probably based on these studies clinicians were administering antibiotics at first (I don’t remember the exact title, but there was a French study on 24 patients where azithromycin was suggested to improve clinical picture.

I would suggest to place Material and Methods after the Introduction -> this will provide more clarity

Material and methods

Why did you include only patients with the first COVID-19 diagnosis?

line 211: Please name all comorbidities instead of etc. 

Treatment -> I would suggest to show it in a Table it would provide more clarity

lines 236-7: Please explain how you compared quantitative data with the chi2 test. I believe this sentence should be rephrased.

Results

line 73: the second range after providing interquartile range is not necessary in my opinion, you may just provide the IQR

Discussion

I would suggest adding some insight on usage of tocilizumab and antibiotics as preventive measures against super bacterial infections.

Author Response

January 20 of 2023

Manuscript ID: antibiotics-2130321

Title: Use of systemic antibiotics in patients with COVID-19 in Colombia. A

cross-sectional study

Dear editors

Antibiotics

Thank you very much for sending us your comments and allowing us to make corrections. We sent the manuscript and the details of each adjustment that was made following the recommendations made by the evaluators. We are sure that they improve the quality of the manuscript.

Antibiotics

Reviewer 1

Comments and Suggestions for Authors

Thank you for inviting me to review this interesting article on antibiotics use in patients with COVID-19. It is well written and in an interesting way, however, there are some remarks I would suggest implementing in order to increase it scientific value:

Abstract

The Abstract contains 330 words -> Please see the Instructions for the authors, the abstract should not be longer than 200 words

Answer: Adjustments are made. The abstract was left with 200 words (without taking into account the keywords).

Introduction

In the introduction section I would suggest adding some information of first studies that showed antibiotic use in COVID-19 necessary. Probably based on these studies clinicians were administering antibiotics at first (I don’t remember the exact title, but there was a French study on 24 patients where azithromycin was suggested to improve clinical picture.

Answer: This study is included in the introduction.

I would suggest to place Material and Methods after the Introduction -> this will provide more clarity

Answer: according to the rules of the journal Antibiotics, it contemplates that after the introduction the results must be located.

Material and methods

Why did you include only patients with the first COVID-19 diagnosis?

Answer: because the intention was to identify the use of antibiotics in those patients who had covid19 upon admission to hospital and determine what was being used as part of their treatment. But it must be taken into account that the main comorbidities that these patients had were considered.

line 211: Please name all comorbidities instead of etc. 

Answer: Adjustment is done.

Treatment -> I would suggest to show it in a Table it would provide more clarity

Answer: This information is found in Supplementary Table 1.

lines 236-7: Please explain how you compared quantitative data with the chi2 test. I believe this sentence should be rephrased.

Response: The statistical analysis section is redrafted.

Results

line 73: the second range after providing interquartile range is not necessary in my opinion, you may just provide the IQR

Answer: Adjustment is done.

Discussion

I would suggest adding some insight on usage of tocilizumab and antibiotics as preventive measures against super bacterial infections.

Response: The data was verified and no patient received tocilizumab at the hospital level, therefore, it is not a possible explanation for the use of antibiotics in this group of patients, so we consider that it is not necessary to include this topic in the discussion.

                The authors

Reviewer 2 Report

We recommend small revisions to the authors when editing the text, as follows>

- line 18 – correct form – between 2020-2022

- line 22 – correct form – were male.

- in the main article – briefly describe Watch group, Access antibiotics

- lines 68-69 – correct form – between 2020-2022

- line 72 – correct form – were male

- Table 1 – correct form – fist column – Male

- line 125 – correct form – of male

- line 130 – correct form – some systemic antibiotics

- for References: mark the volume number with italics and the section references between round brackets, also italics; also, mark reference pages for some articles as follows:

ref 1 - 91(1); ref 6 – 25; ref 7 – 8(3); ref 8 – 71(9); ref 9 – 17(8); ref 10 – 27(4); ref 11 – 20(5); ref 12 – 81(2); ref 14 – 24(4); ref 15 – 11(3), 333; ref 16 - 11(6), 764; ref 17 – 16(11); ref 18 – 11(6); ref 19 – 11(6), 738; ref 20 – 65(11); ref 21 – 21(1); ref 22 – 120(9); ref 23 – 11(2), 176; ref 24 – 116(7); ref 25 – 81(6); ref 26 – 43(8); ref 27 – 16(5); ref 30 – 19(19), 11931; ref 31 – 9(4); ref 32 – 11, 1198.

Author Response

January 20 of 2023

Manuscript ID: antibiotics-2130321

Title: Use of systemic antibiotics in patients with COVID-19 in Colombia. A

cross-sectional study

Dear editors

Antibiotics

Thank you very much for sending us your comments and allowing us to make corrections. We sent the manuscript and the details of each adjustment that was made following the recommendations made by the evaluators. We are sure that they improve the quality of the manuscript.

Reviewer 2

Comments and Suggestions for Authors

We recommend small revisions to the authors when editing the text, as follows>

- line 18 – correct form – between 2020-2022

- line 22 – correct form – were male.

- in the main article – briefly describe Watch group, Access antibiotics

- lines 68-69 – correct form – between 2020-2022

- line 72 – correct form – were male

- Table 1 – correct form – fist column – Male

- line 125 – correct form – of male

- line 130 – correct form – some systemic antibiotics

Answer: All suggested settings are made. A paragraph is added in the introduction about the AWaRe strategy.

- for References: mark the volume number with italics and the section references between round brackets, also italics; also, mark reference pages for some articles as follows:

ref 1 - 91(1); ref 6 – 25; ref 7 – 8(3); ref 8 – 71(9); ref 9 – 17(8); ref 10 – 27(4); ref 11 – 20(5); ref 12 – 81(2); ref 14 – 24(4); ref 15 – 11(3), 333; ref 16 - 11(6), 764; ref 17 – 16(11); ref 18 – 11(6); ref 19 – 11(6), 738; ref 20 – 65(11); ref 21 – 21(1); ref 22 – 120(9); ref 23 – 11(2), 176; ref 24 – 116(7); ref 25 – 81(6); ref 26 – 43(8); ref 27 – 16(5); ref 30 – 19(19), 11931; ref 31 – 9(4); ref 32 – 11, 1198.

Answer: All settings are done.

The authors

Reviewer 3 Report

Valladales-Restrepo et al., surveyed antibiotic usage with SARS-CoV-2 patients in Columbia. This study is a well fit for the journal, well-designed, educational, and timely, and provides a large sample size (n>10,000) including minors and elderly across 8 clinics and 181 cities. Largely, the Result/Discussion needs major revise. Specific comments are listed below.

1. It would be helpful to have a map of the regions to explain the pacific, central, Caribbean, etc. Especially it is not clear how 8 clinics involve 181 cities.

2. Table1 horizontal line is needed to divide each category

3. More than half of patients (57%) ended up with antibiotics. What determined the antibiotic intake? Is any reason other than what the authors described in lines 49-50? If so, any evidence of bacterial infection the authors can provide? How is this compared to non-SARS-CoV-2 infection patients?

4. Lines 110-115: please revise this sentence as it is a very long single sentence.

5. section 2.2 multivariate analysis is the most informative data from this study yet the authors have only 2 sentences to explain this section. I recommend the authors explain and expand this section fully and comprehensively.

6. table3: it is surprising diabetes mellitus, ischemic heart disease, and COPD didn't show significance. Any explanation for this? Also which reference should be looked at for emergency care?

7. line 125: authors mentioned male predominance was observed. Are there statistics the authors can provide? For emergency care, it is actually less than 50% according to table 1.

8. line 127: diabetes mellitus does not show statistical significance according to table 3. How would the authors reach this conclusion?

9. The Discussion section: I find this section the most problematic. First of all, it reads quite strangely when the authors put other studies in the center instead of the current study. For example, for lines 132-133,  it should be written "the prevalence of antibiotic use found in our study (57%) was lower than the US (77%)" as this is a more current study-centered sentence. It feels backward right now how the authors wrote the Discussion section. Also, I found the Discussion section disconnected from the results as if this section was written prior. The Discussion section should be the area the authors dive deep into to connect their findings (from the result section) with what we know from other studies.

10. Conclusion section: can authors provide some detailed future direction or recommendations other than general sentences like lines 248-250?

Author Response

January 20 of 2023

Manuscript ID: antibiotics-2130321

Title: Use of systemic antibiotics in patients with COVID-19 in Colombia. A

cross-sectional study

Dear editors

Antibiotics

Thank you very much for sending us your comments and allowing us to make corrections. We sent the manuscript and the details of each adjustment that was made following the recommendations made by the evaluators. We are sure that they improve the quality of the manuscript.

Reviewer 3

Comments and Suggestions for Authors

Valladales-Restrepo et al., surveyed antibiotic usage with SARS-CoV-2 patients in Columbia. This study is a well fit for the journal, well-designed, educational, and timely, and provides a large sample size (n>10,000) including minors and elderly across 8 clinics and 181 cities. Largely, the Result/Discussion needs major revise. Specific comments are listed below.

  1. It would be helpful to have a map of the regions to explain the pacific, central, Caribbean, etc. Especially it is not clear how 8 clinics involve 181 cities.

Answer: These are highly complex clinics that are reference centers and also treat patients referred from other cities.

  1. Table1 horizontal line is needed to divide each category.

Answer: according to the rules of the journal, the tables do not have horizontal lines inside.

  1. More than half of patients (57%) ended up with antibiotics. What determined the antibiotic intake? Is any reason other than what the authors described in lines 49-50? If so, any evidence of bacterial infection the authors can provide? How is this compared to non-SARS-CoV-2 infection patients?

Answer: Clarification is done in the methods. The management received by the patients was established from the dispensing of the drugs. The concomitant presence of other infectious pathologies could not be determined (it is added in the limitations of the study). All patients had paraclinical confirmation of SARS-CoV2, so the study does not compare patients without COVID-19.

  1. Lines 110-115: please revise this sentence as it is a very long single sentence.

Answer: Adjustment is done.

  1. section 2.2 multivariate analysis is the most informative data from this study yet the authors have only 2 sentences to explain this section. I recommend the authors explain and expand this section fully and comprehensively.

Answer: we adjust the paragraph and include the OR values to make the reading clearer.

  1. Table 3: it is surprising diabetes mellitus, ischemic heart disease, and COPD didn't show significance. Any explanation for this? Also which reference should be looked at for emergency care?

Answer: This aspect is added in the 4th paragraph of the discussion.  References 32 and 33.

  1. line 125: authors mentioned male predominance was observed. Are there statistics the authors can provide? For emergency care, it is actually less than 50% according to table 1.

Answer: The first line of the results mentions a global predominance of men (56.4%; n=6155). In addition, the discussion sentence is adjusted: … and there was a global predominance of male, as described in other studies (51.5-69.0%). References: 10,17,18,20-24).

  1. line 127: diabetes mellitus does not show statistical significance according to table 3. How would the authors reach this conclusion?

This aspect is added in the 4th paragraph of the discussion.  References 32 and 33.

  1. The Discussion section: I find this section the most problematic. First of all, it reads quite strangely when the authors put other studies in the center instead of the current study. For example, for lines 132-133, it should be written "the prevalence of antibiotic use found in our study (57%) was lower than the US (77%)" as this is a more current study-centered sentence. It feels backward right now how the authors wrote the Discussion section. Also, I found the Discussion section disconnected from the results as if this section was written prior. The Discussion section should be the area the authors dive deep into to connect their findings (from the result section) with what we know from other studies.

Answer: Thank you for your comment. In the discussion we clearly compared our findings with those of other studies, including the prevalence of antibiotic use, which was a global problem and not only of this analysis (References 18, 20, 21, 22, 23, 25, 26, 27). It was located in the context of the AWaRe strategy proposed by the World Health Organization to promote the adequate use of antibiotics (references 20,21,25,26, 30), and the variables that were associated with the use of antibiotics were addressed. (references 11, 18, 23, 27, 32, 33, 34, 35).

  1. Conclusion section: can authors provide some detailed future direction or recommendations other than general sentences like lines 248-250?

Answer: Adjustment is done.

The authors

Round 2

Reviewer 3 Report

No further change is needed.